# Airborne observations reveal the fate of the methane from the Nord Stream pipelines

Friedemann Reum [1] ✉, Julia Marshall [1], Henry C. Bittig [2], Lutz Bretschneider[3], Göran Broström[4], Anusha L. Dissanayake [5], Theo Glauch [1,6], Klaus-Dirk Gottschaldt [1], Jonas Gros [7], Heidi Huntrieser [1], Astrid Lampert [3], Michael Lichtenstern[1], Scot M. Miller [8], Martin Mohrmann [4,9], Falk Pätzold [3], Magdalena Pühl[1], Gregor Rehder [2] & Anke Roiger [1]

The Nord Stream pipeline leaks on 26 September 2022 released 465 ± 20 kt of methane into the atmosphere, which is the largest recorded transient anthropogenic methane emission event. While most of the gas escaped directly to the atmosphere, a fraction dissolved in the water. So far, studies on the fate of this dissolved methane rely on pipeline volumetric estimates or spatially sparse concentration measurements and ocean models. Here, we use atmospheric measurements with broad spatial coverage obtained from an airborne platform to estimate outgassing of 19-48 t h$^{-1}$ on 5 October 2022. Our results broadly agree with ocean models but reveal uncertainties such as inaccuracies in their spatial emission distribution. Thus, we provide a data-driven constraint on the fate of the methane from the Nord Stream pipelines in the Baltic Sea. These results demonstrate the benefit of a fast-response air-borne mission to track a dynamic methane emission event.

Methane is the second-most important anthropogenic greenhouse gas in the atmosphere today[1]. One important source is the supply chain of natural gas[2], which consists primarily of methane [e.g. ref. 3]. Inventory data put recent emissions from the largest natural gas supply chains at 26 Tg $CH_4$ yr$^{-1}$ globally[4] and total emissions related to oil and gas exploitation at 80 Tg $CH_4$ yr$^{-1}$, representing 63% of anthropogenic methane emissions. These estimates have large uncertainties[2,4]. Atmospheric observations have revealed that emissions in inventories can be underestimated because few hot spots can dominate regional budgets [e.g. ref. 5].

An extraordinary natural gas emission event occurred on 26 September 2022 in the Baltic Sea, when explosions ruptured the Nord Stream 1 and 2 pipelines at multiple locations. The locations are shown in Fig. 1. The first explosion took place at 00:03 UTC[6] and damaged

pipe A of the Nord Stream 2 pipeline (hereafter referred to as NS2A) southeast of Bornholm at 54.88°N, 15.41°E[7] at a depth of about 70 m[8]. At 17:03 UTC[6], multiple explosions[9] occurred northeast of Bornholm and damaged both pipes of Nord Stream 1 (NS1A: 55.56°N, 15.79°E[10], NS1B: 55.54°N, 15.70°E[11]), and NS2A (55.54°N, 15.78°E)[10] at a depth of about 75 m[8]. These northern leaks were less than 6.5 km apart.

At the time, none of the pipelines were in operation, but none-theless were filled with pressurised natural gas[12]. The total amount of methane vented into the atmosphere is estimated to be 465 ± 20 kt[13]. This amount is equal to ~ 30 % of the annual anthropogenic methane emissions from Germany[14]. It far exceeds other exceptional transient anthropogenic methane emission events that have been quantified using atmospheric observations from aircraft and space, such as well blowouts in Aliso Canyon (up to 60 t h$^{-1}$ and 97.1 kt total)[15], Louisiana

[1]Deutsches Zentrum für Luft- und Raumfahrt e.V., Institut für Physik der Atmosphäre, Oberpfaffenhofen, Germany. [2]Leibniz Institute for Baltic Sea Research Warnemünde, Rostock, Germany. [3]Technische Universität Braunschweig, Institute of Flight Guidance, Braunschweig, Germany. [4]University of Gothenburg, Department of Marine Sciences, Gothenburg, Sweden. [5]Independent Researcher, EnvSoln, Badulla, Uva Province, Sri Lanka. [6]University of Heidelberg, Institute of Environmental Physics, Heidelberg, Germany. [7]Independent Researcher, Villars-sur-Glâne, Switzerland. [8]Johns Hopkins University, Baltimore, MD, USA. [9]Voice of the Ocean, Västra Frölunda, Sweden. ✉e-mail: friedemann.reum@dlr.de

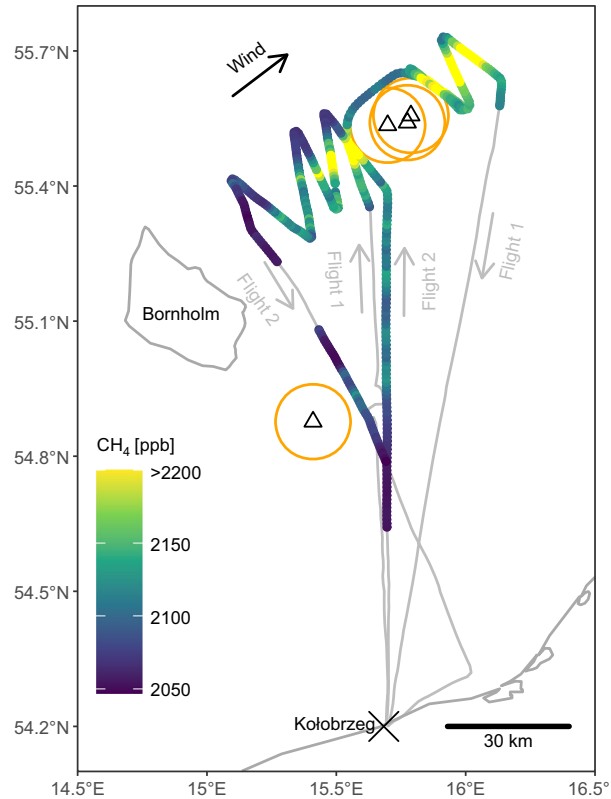

**Fig. 1 | Overview of the study area and atmospheric methane measurements.** Paths of the two flights are in grey, with leak locations as black triangles, flight exclusion zones as orange circles and the black arrow indicating the average wind direction (217°). The parts of the flights within the boundary layer, i.e. used for emission estimation, are coloured according to the observed methane mole fractions. The colour scale cuts off the highest methane peaks in order to better visualise variability in sections with lower methane mole fractions.

(up to 101 (49–127) t h⁻¹ and 49 (21–63) kt total)[16], Ohio (120 ± 32 t h⁻¹ and 60 ± 15 kt total)[17], Texas (up to 27.6 ± 8.8 t h⁻¹ and 4.8 ± 1 kt total)[18] and the Elgin platform release (up to 4.7 ± 0.7 t h⁻¹)[19].

The gas bubbles reached the water surface, creating surface plume expressions several hundred metres in diameter [e.g. ref. 20]. The bubbling ceased by 1 October 2022 at the southern leak site[21] and by 3 October at the NS1A leak site[10]. At the northern NS2A leak site, a small surface plume expression of ~15–25 m was observed until 4 October[10,22].

Methane emissions from the Nord Stream event have been observed in the atmosphere from multiple platforms, including in situ observations from the Integrated Carbon Observation System (ICOS) network of greenhouse gas observing stations[23–26] and via remote sensing from satellites[27–29]. An overview can be found in ref. 13.

The vast majority of the methane from the pipelines was vented directly to the atmosphere via bubble transport. However, a portion dissolved in the water of the Baltic Sea during upward migration through the water column [e.g. ref. 20]. This process was documented by measurements of the concentration of dissolved methane, which was enhanced over background values by several orders of magnitude around the leak sites in early October[30]. The total amount of methane that dissolved was 9–15 kt[13].

Once dissolved, methane is transported by ocean currents. In the Nord Stream case, vertical mixing was fast in the mixed layer due to wind-induced turbulence, as indicated by homogeneous concentrations of dissolved methane in the mixed layer without consistent depth gradients[30]. Mixing with deeper water masses, i.e. below the thermocline (at the time at ~30 m depth[30]), was slower and occurred primarily

via entrainment when the mixed layer deepened in the months following the Nord Stream explosions[30].

Methane is removed from seawater by outgassing to the atmosphere[31] and microbial oxidation. In the first days after the pipeline explosions, outgassing likely dominated because microbial methane oxidation typically takes months in the Baltic Sea[32,33], and the timescale for methanotrophic microbial communities to adapt to a massively increased methane availability is on the same order[34]. In the Bornholm Basin, natural microbial methane oxidation is likely restricted to below the halocline (at the time at ~50 m depth[30]), since methane from the sediment is trapped there and forms the basis for methanotrophic communities[35,36]. By contrast, the e-folding time for venting methane from the mixed layer to the atmosphere at the leak sites during the Nord Stream event was about five days (based on our ocean model for a mixed layer depth of ~20 m, see Supplementary Fig. 11, 12). Therefore, in the first few weeks after dissolution from the bubbles, venting from the mixed layer to the atmosphere was likely the dominant methane removal process.

Recent studies have used estimates of the amount of methane released from the pipes into the water[20] and observations of dissolved methane in the water[30,37] to investigate the amount of methane that dissolved at the Nord Stream leak sites and its fate. However, the spatial coverage of these datasets is sparse, and emission estimation relies on modelling of bubble plumes and oceanic methane transport, which introduces uncertainties.

In this work, we estimate the outgassing of dissolved methane based on atmospheric observations with extensive spatial coverage, providing a spatially resolved snapshot of the emissions. We obtained the observations using an airborne platform on 5 October 2022. Using a model of atmospheric transport and an inverse estimation technique, we estimate the spatial pattern and magnitude of methane emissions to the atmosphere on that day. We compare these findings to emission estimates that are based on models of methane dissolution, transport and outgassing. The models broadly agree with the atmospheric estimate, but the comparison also reveals potential uncertainties in both emission estimation approaches. Thus, our work provides insight into the fate of the methane from the Nord Stream pipes via an independent estimate of the emissions.

## Results
### Atmospheric methane observations
We observed atmospheric methane mole fractions around the Nord Stream leak sites during two research flights of 3 h each on 5 October 2022. The measurement platform was the drag sonde HELiPOD[38], which was attached as a sling load to a helicopter. Atmospheric methane mole fractions were measured using a cavity ring-down spectrometer (Picarro G2401-m).

The main objective of the first flight was to sample the methane background mole fraction during one flight leg upwind (southwest) of the northern leak sites, and enhancements during multiple flight legs downwind (northeast) of them. Since we observed the largest methane enhancements during the upwind flight leg, we adapted the path of the second flight at short notice to study the methane distribution upwind of the northern leak sites. We aimed to spend as much flight time as possible in the planetary boundary layer (here, below 300–600 m) since it is directly influenced by surface fluxes. We also wanted to sample as close as possible to the leak sites. These objectives were impaired by temporary flight restrictions, i.e. by limiting us to flight altitudes above the boundary layer during the transfer flight legs of flight 1, and not allowing entry into flight exclusion zones of 5 nautical miles (about 9 km) radius up to 3500 ft (about 1 km) altitude around the leak sites[39–41], as shown in Fig. 1. These flight exclusion zones were similar to the one that had been in place during the flights that were undertaken to quantify methane emissions from the Elgin platform release in 2018[19].

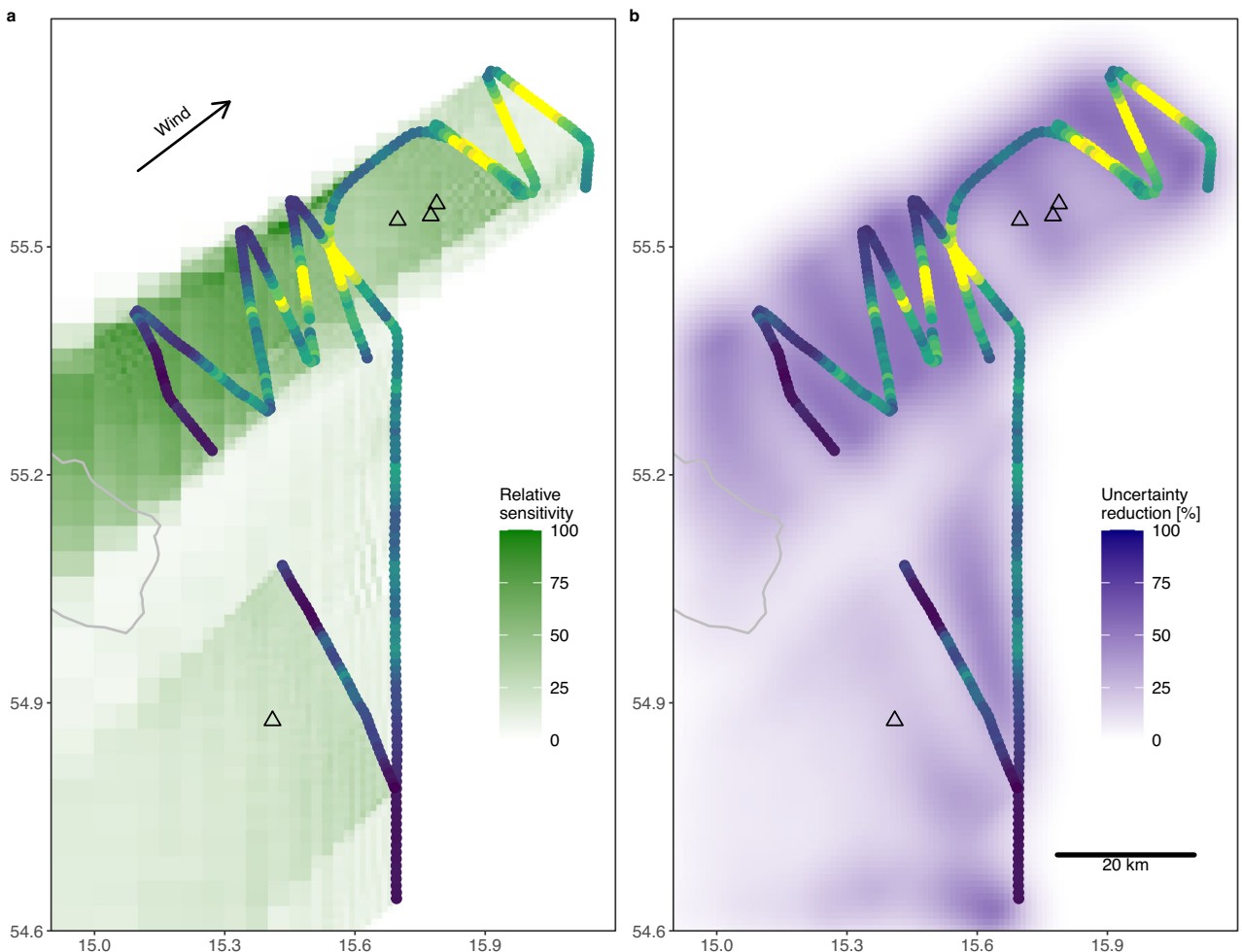

**Fig. 2 | Spatial distributions of the sensitivity of the observations to surface fluxes (footprints) and reduction of random uncertainty. a** Sum of footprints (surface influence functions) of all observations, indicating the area to which the observations are sensitive, **b** reduction of emission uncertainty. Shown here are values for Meteo A and BG $CO_2$ (see Methods for definitions); results for other settings are similar. The color scale of the superimposed observations is the same as in Fig. 1.

The boundary layer observations are shown in Fig. 1. During both flights, we observed the highest methane mole fractions up- and downwind of the northern leak sites (20-s average up to 2380 ppb; rounded to 10 ppb), as a series of peaks that lined up roughly along the wind direction. They gradually diminished in the upwind direction, indicating an underlying area source. About 45 km upwind of the northern leak sites, no distinct peak was observed anymore (2050 ppb), which indicates that our flight path covered the spatial extent of the emitting area in the upwind direction. Downwind of the northern leak sites, the methane mole fractions did not increase further, which indicates that we also covered the emitting area in the downwind direction. Methane mole fractions downwind of the southern leak site were much lower than in the north, with a maximum of 2130 ppb. In the area between the leak sites, there was a methane gradient along the wind direction from 2070 ppb to 2090 ppb (respective averages of the two boundary layer flight legs in that area).

In general, the edges of our flight pattern featured the lowest observed methane values (except for the downwind (northeast) edges, because they are influenced by the emissions we estimate). As pointed out above, this distribution indicates that our flight paths encompassed the vast majority of the emitting area. However, the lowest methane concentrations at the northern edge were up to 50 ppb higher than the lowest ones during the campaign, which were observed in the west (i.e. upwind) of the northern leaks and south of

the southern leak. This is a large gradient for background mole fractions. Therefore, these observations may have been influenced by emissions in the area of interest and not represent background conditions. This ambiguity causes uncertainty in our emission estimate, which we address by computing emissions with both a lower and an upper limit for the background mole fractions (see Methods and Supplementary Note 1).

Overall, the widespread enhancements, in particular the peaks upwind of the northern leaks, cannot be explained by point sources at the four leak locations. Instead, the observations indicate a source distributed over several tens of kilometres around all leak sites.

## Methane emissions estimated from atmospheric data

We retrieve methane emissions from the atmospheric data via an inverse model of atmospheric transport (see Methods). Owing to the wind direction, the observations were sensitive to emissions southwest of the flight tracks, as indicated by their footprints and the reduction of flux uncertainty (Fig. 2). In total, the estimated methane source strength is 19–48 t h$^{-1}$, originating from a spatially distributed source around the Nord Stream leaks. This estimate is representative of 5 October 2022, 8:00–15:30 UTC, which is the period to which our observations are sensitive (based on the period of data collection and the atmospheric flushing time of the domain).

The range of 29 t h$^{-1}$ in the total emission estimate stems from modifications of the atmospheric model setup, which we use to quantify systematic uncertainties in the methane background (lower and upper limit) and the atmospheric transport model (two meteorological simulations and two transport schemes), for a total of eight individual emission estimates (see Methods and Supplementary Note 3). The Bayesian random uncertainty is 5 t h$^{-1}$ for the lower bound and 8 t h$^{-1}$ for the upper bound, smaller than the systematic uncertainties. The uncertainty in the total emissions is due mostly to uncertainty in the boundary layer height and in the methane background mole fractions, which we describe in the following paragraphs.

The ambiguity in the methane background (Section "Atmospheric methane observations") requires relying on additional data sources and assumptions. In particular, the upper bound of the methane background, which determines the lower bound of the emission estimates, relies on the observation that $CO_2$ and methane backgrounds could have been strongly correlated during our measurement campaign (e.g. due to entrainment from the free troposphere or long-range transport of accumulations). This assumption is uncertain since $CO_2$ and methane have different sources and sinks, and therefore, we use it only to derive a lower bound of emissions (see Methods for details on the methane background estimation).

The most important aspects of the meteorological simulations for an accurate emission estimate are the modelled winds and boundary layer heights. Modelled wind speeds agree with observations within measured variability (Supplementary Fig. 7). The modelled wind directions exhibit small biases (Supplementary Note 2.1 and Supplementary Fig. 6), but they do not have a large impact on the retrieved spatial emission pattern (Supplementary Fig. 9). There is a mean difference in the boundary layer heights between the two meteorological simulations. However, in most cases, both simulations agree with pronounced capping inversions that we observed in the morning (flight 1), making it difficult to differentiate the accuracy of the boundary layer heights of the two simulations despite their mean difference (Supplementary Note 2.2). Based on our analyses, we do not consider any of the setup modifications to be conclusively more likely than the others, which contributes to the emission uncertainty (Supplementary Note 3).

The spatial distribution of retrieved methane emissions is shown in Fig. 3a. Most aspects of the pattern are robust against the variations in the model setup that result in uncertainty in the total emission estimate (Supplementary Fig. 9). The most prominent emission feature is a band of emissions that extends from the northern leak locations to the southwest, tracking the highest atmospheric methane enhancements observed during flight 2. The emission band turns southeast between the two westernmost flight legs. Little emissions are retrieved further to the west (i.e. upwind) since only comparatively small enhancements were observed during the westernmost flight leg. An additional, isolated emission hot spot is placed 10 km north of the leak sites by the model. It could be related to the larger emission band but disconnected due to spatial data sparsity. Note that ship-borne measurements revealed a hot spot of dissolved methane close-by in the period 3–6 October 2022[37].

The inversions indicate that the area around the southern leak was emitting methane as well, albeit with lower emission rates than in the north. In the area between the southern leak and the densely sampled corridor around the northern leaks, sampling in the planetary boundary layer was limited to two flight legs, and thus was much sparser than in the north. Since methane was enhanced along most of the eastern of these two flight legs, the inversions predict that the area upwind is emitting methane. In contrast, little emissions are retrieved upwind of the western flight leg. Except for the area around the southern leak site and a gap in the data (when we briefly left the boundary layer to determine the boundary layer height), this flight leg appears to constitute a western boundary of the emitting area. The sum of the estimated emissions from this area is, in first order, determined by the gradient between the observations and background. However, unlike in the densely sampled corridor around the northern leaks, the spatial pattern of the retrieved emissions has little variability and is dominated by the locations of the flight paths. Thus, while the measurements allow constraining the total emissions in the southern part of the domain, the sampling was, unlike in the north, too sparse to retrieve potential details in the spatial structure of the emissions there.

## Comparison to ocean models

The main feature in the emissions retrieved from the atmospheric data, i.e. the emission band that extends from the northern leak sites to the southwest (Fig. 3), roughly corresponds in direction to the main oceanic circulation in the Bornholm Basin, a topography-driven cyclonic gyre[42] (the mean surface currents from 26 September to 5 October 2022 are shown in Supplementary Fig. 14). To confirm whether the emissions we estimate originate from the Nord Stream leaks, we developed a model to simulate the fate of methane that dissolved at the leak sites. The model (henceforth simply referred to as the ocean model) consists of the near-field model of the bubble plumes from ref. 20 coupled to a simple Lagrangian tracer dispersion and outgassing model developed for this study (see Methods and Supplementary Methods 3.1). According to the near-field model, 10.8 kt of methane dissolved from the bubble plumes above the leak sites, and 98.5% of it was channelled to the mixed layer[20]. The Lagrangian tracer dispersion model then simulates diffusive and advective transport with ocean currents, as well as outgassing of the dissolved methane to the atmosphere. Owing to the focus of this study on the first 9 days of transport modelling (from the explosions on 26 September until our atmospheric measurements on 5 October), we simplify the vertical structure of the model by assuming that 100% of dissolved methane is in the mixed layer, vertical mixing therein is instantaneous, and that there is no mixing with water masses below. These simplifications effectively reduce the oceanic dispersion model to a 2D model of the mixed layer. Given the short timescale and focus on the mixed layer, the model ignores microbial oxidation (cf. Introduction). The simplification to 2D would not be suitable for longer simulations, but a comparison to the Lagrangian transport and outgassing model from ref. 20 shows that the simplification has little impact on concentrations in the mixed layer and thus emissions on 5 October (Supplementary Fig. 11), and is therefore suitable for our purposes. For additional details about the ocean model, see Methods and Supplementary Methods 3.1.

On 5 October 2022, 8:00–15:30 UTC, the ocean model yields methane emissions of 32 t h$^{-1}$. This estimate falls in the range of the emissions obtained from the atmospheric data (19–48 t h$^{-1}$). The size of the emitting area (Table 1) and the broad strokes of the spatial distribution agree with the inversion results: at the northern leaks, the ocean model predicts a similar characteristic band of emissions, although it appears rotated such that the main direction is not southwest, but west (Fig. 3b). At the southern leak, the ocean model predicts lower emissions than in the north (owing to the lower amount of methane that dissolved there[20]), which is also the case in the inverse models. Between the southern and northern leak sites, the ocean model predicts a gap in the emission distribution, while the inversions show a broad source between the boundary layer transfer flights. Overall, the ocean model corroborates that the emissions retrieved using the atmospheric data originated from the outgassing of methane that had dissolved at the Nord Stream leak sites.

Despite the large-scale similarities, the spatial distribution of emissions from the ocean model has a low correlation with the atmospheric estimate (0.3; Table 1). As a consequence, applying the atmospheric transport model to the ocean model emissions does not simulate the atmospheric data accurately (Fig. 3d and Table 1). The

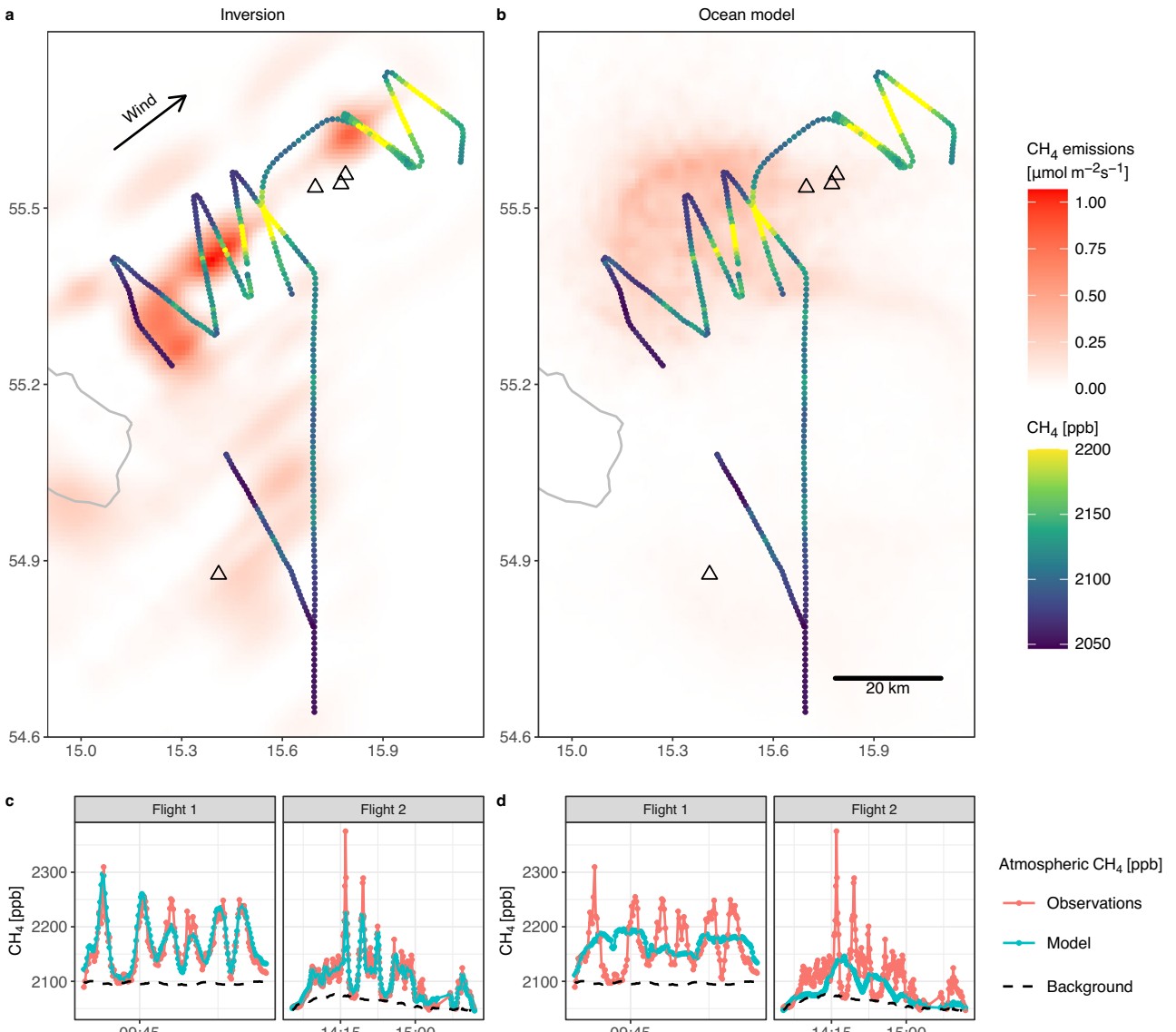

**Fig. 3 | Modelled methane emissions and atmospheric methane mole fractions.**
Top row: Maps of methane emissions retrieved from the atmospheric data via inverse modelling (**a**) and modelled by the ocean model (**b**). The black triangles mark the leak locations. Bottom row: Time series of atmospheric methane mole fractions, observed and modelled using the inversion result (**c**) and the ocean model (**d**). The inverse model results shown in (**a**, **c**) are from one of four inversions, which yields emissions of 34 t h⁻¹, which is close to the centre of all estimates. See Supplementary Note 3 for details and results obtained from all four inversions.

discrepancies, particularly in the north where sampling was dense, are likely due to inaccuracies in the underlying ocean currents, since the small domain and agreement of modelled and observed winds makes large errors in the spatial mapping of emissions in the atmospheric inverse model unlikely. By contrast, the ocean model relies on transport over the 9 days since the leaks started, allowing inaccuracies in the underlying ocean currents to accumulate. The exact position and strength of the Bornholm Basin gyre is wind-dependent and can fluctuate over short timescales[42], which likely explains the displacement of the plume of dissolved methane revealed by the atmospheric data, i.e. the band of emissions that extended from the northern leak sites to the southwest. Note that despite inaccuracies in the short term, gyre circulation modelled based on the surface currents we use[43] is able to reproduce the observed broader plume of dissolved methane over longer timescales and larger spatial scales[30].

In several other studies, ocean models have been developed to study the fate of dissolved methane from the Nord Stream leaks[20,30,37]. Here, we compare our results to those from the ocean model developed in ref. 30, hereafter called M25. Many aspects of M25 are similar

to our ocean model, including the Lagrangian modelling approach, the total amount of dissolved methane, the underlying ocean currents and the atmospheric wind speed (Supplementary Table 3). However, unlike our ocean model, M25 starts with a vertically homogeneous distribution of dissolved methane throughout the water column, accounts for vertical transport and uses a different outgassing model. A brief model description for M25 is given in Supplementary Methods 3.2, with more details in ref. 30.

The spatial emission distribution in model M25 is broadly similar to the one obtained by our model (Supplementary Fig. 10), owing to its use of the same ocean current product for lateral transport. Consequently, M25 does not match the exact spatial distribution of inversion-optimised emissions and the atmospheric data well either (Supplementary Tables 5–6 and Supplementary Fig. 10). The total emissions for 5 October are lower in M25 (13 (11–18) t h⁻¹) than in our ocean model (32 t h⁻¹). Since both ocean models are similar in terms of the total amount of methane that dissolved at the leaks and the timing of the dissolution (Supplementary Table 3 and Supplementary Fig. 12), the reasons for the different total emissions are the vertical

**Table 1 | Summary of methane emissions on 5 October 2022 from the ocean model in comparison with inversion results and the atmospheric data used in the inversions**

| Metric | Inversion | Ocean model |
|---|---|---|
| Emissions | | |
| Total emissions [t h⁻¹] | 19–48 | 32 |
| Correlation with emissions from inversion | 0.79–1 | 0.25–0.34 |
| Area with 90 % of total emissions [10⁹ m²] | 3.2–3.7 | 4.7 |
| Modelled atmospheric methane mole fractions | | |
| Correlation with atmospheric data | 0.92–0.95 | 0.41–0.64 |
| Mean bias [ppb] | −0.7–0.4 | −33-(-1) |
| Mean enhancement [ppb] | 51–70 | 38–50 |

The ranges represent the variations in the inverse model setup used to assess systematic uncertainties, i.e. background mole fractions, meteorological fields, and, for the total emissions, the transport model (see Methods and Supplementary Notes 1–3 for details). The correlations are Pearson correlation coefficients.

distribution of dissolved methane and the gas transfer velocity. Since some dissolved methane is below the mixed layer in M25, the dissolved methane concentration in the mixed layer is initially lower than in our ocean model, which decreases outgassing. On the other hand, gas transfer velocities are higher in M25 owing to a different parameterisation (Supplementary Fig. 13), and methane-rich water from below is entrained into the mixed layer, which increases outgassing. The net effect is that the dissolved methane vents more slowly to the atmosphere in M25 than in our ocean model (Supplementary Fig. 12), and on 5 October, the concentrations at the surface are higher in our ocean model than in M25 (Supplementary Fig. 11), which explains the higher emissions on that day. The total emission estimate from M25 is at the lower end of the atmospheric estimate, while the emission estimate of our ocean model falls in the middle. Thus, the atmospheric result indicates that more dissolved methane may have been present in the mixed layer than in M25 on 5 October. Possible explanations include an underestimation of the fraction that reached the mixed layer (as indicated by the near-field model from ref. 20), overestimation of the gas transfer velocity (leading to higher outgassing rates prior to and hence lower concentrations in the water on the day of our observations) or an underestimation of the total amount of dissolved methane (which dominates the uncertainty range of M25 on 5 October[30]).

## Discussion

On 5 October 2022, we observed atmospheric methane mole fractions from an airborne platform around the locations of the Nord Stream leaks that started on 26 September 2022, the largest ever recorded transient anthropogenic methane emission event. Emissions retrieved from these atmospheric data via inverse modelling reveal a spatially distributed source of 19–48 t h⁻¹. The most prominent emission feature is a band of emissions that extends from the northern leak sites to the southwest to a distance of 45 km from the leaks, and then turns to the southeast. The direction of the emission band roughly corresponds to the long-term mean oceanic circulation, i.e. the Bornholm Basin gyre (ref. 42 and Supplementary Fig. 14). In other parts of the domain, the spatial sampling was not dense enough to retrieve potential details in the spatial structure in the emissions.

The range in the total emissions, 19–48 t h⁻¹, represents the lower and the upper bound of eight individual emission estimates that differ in the atmospheric transport model and atmospheric methane background, representing uncertainties in these model components. We choose this approach to quantify systematic uncertainties that are often ignored in atmospheric inverse modelling studies and which

dominate over random uncertainty in our case. The often-used Bayesian posterior uncertainty, which quantifies random uncertainties, is much lower in our case. The systematic uncertainty is due in roughly equal parts to atmospheric transport (most importantly, boundary layer height) and the methane background. While our analysis provides some insight into inverse model uncertainty, it is based on a few changes in the model setup and, therefore, does not quantify the full range of transport model uncertainty. An additional uncertainty is that we might have missed potential emissions outside of the area to which our observations are sensitive, which could be the case to the east (i.e. downwind) or north of our flight path (Fig. 2). Note that the recent study by Abrahamsson et al. indicates that emission rates north of our flight paths were low compared to those in the area we sampled[37].

An ocean model developed for this study confirms that the spatial distribution of emissions obtained from the atmospheric data can be explained by the outgassing of methane that dissolved from the bubble plumes from the Nord Stream leaks, and was then transported with ocean currents. Details in the spatial distribution differ. Most prominently, the band of high emissions that extends from the northern leak sites appears rotated in the ocean model. The likely reason is that inaccuracies in the ocean currents underlying the ocean model accumulated between the beginning of the bubble plumes on 26 September and our measurements on 5 October, and that the atmospheric observations reveal the actual placement of the Bornholm Basin gyre close to the northern leaks during that period.

The atmospheric emission estimate confirms the rough spatial emission distribution obtained with the ocean model from ref. 30 (M25), although the exact spatial distribution differs, for similar reasons as for our ocean model. The estimate of the total emissions on 5 October 2022 by M25 (13 (11–18) t h⁻¹) is at the low end of the atmospheric estimate when taking into account the random uncertainty of its lower bound (19 ± 5 t h⁻¹). The discrepancy of the respective central estimates could be due to factors such as uncertainty in the atmospheric transport model (Section "Methane emissions estimate from atmospheric data") and in the total amount of dissolved methane, in the vertical distribution of dissolved methane and the gas transfer velocity (Section "Comparison to ocean models"). Overall, the emission results obtained from our atmospheric dataset and the oceanic datasets from ref. 30 broadly agree, and the comparison offers insights into potential uncertainties in the methods of both emission estimation approaches.

While the emission rate that originated from dissolved methane on 5 October 2022 was small in comparison to the first days of direct emissions from bubbles[13], it was on the order of magnitude of the emission rate of the Upper Silesian Coal Basin, which, with emissions of ~50 t h⁻¹, is the largest anthropogenic source of methane in Europe[44].

Typically, planning international airborne atmospheric measurement campaigns starts months or even years in advance. In the accelerated timeline for this mission—being airborne 9 days after the explosions— we committed personnel and funding, prepared and integrated the science payload with the platform, organised logistics (e.g. transport), hired aircraft and pilot, obtained flight permissions and planned the flights. Potential future fast-response missions could be improved in several ways. Additional insights on the origin and fate of leaked gas could be gained by measuring the isotopic signature of methane and the mole fractions of other hydrocarbon compounds[19,45,46]. We performed a second airborne campaign to the Nord Stream leak sites in November 2022, where we measured $\Delta^{13}$C-CH$_4$ online and in air samples. Analysis of these data is pending. A more fine-grained characterisation of the boundary layer height could also have been beneficial. However, the best allocation of limited available flight time has to be decided on a case-by-case basis and balance spatial sampling density, repetition and spatial coverage.

Our results demonstrate the benefit of the spatial coverage that can be achieved by airborne observations in a short period of time for tracking spatially distributed methane emissions. Our fast-response mission to the Nord Stream leak sites allowed us to constrain the magnitude and spatial distribution of outgassing of dissolved methane, which other observation platforms did not allow due to their remote location (tall towers), low sensitivity over water (passive spaceborne remote sounders)[13] or spatial sparsity[30]. Thus, this work highlights the benefits of fast on-site monitoring missions for methane emission detection and quantification.

## Methods

### Airborne in situ observations

We obtained atmospheric methane mixing ratios over the Baltic Sea on 5 October 2022 during two research flights out of Kołobrzeg, Poland. The measurement platform was the drag sonde HELiPOD[38], which was attached as a sling load to a helicopter. The helicopter (type Airbus H125, operated by the company Helipoland) enabled a maximum airborne time of 3 h per flight at a typical flight speed of ~40–50 m/s. The HELiPOD was instrumented with a suite of instruments for meteorological conditions, position and attitude (Supplementary Methods 1.1). Atmospheric methane and $CO_2$ mole fractions were measured using a cavity ring-down spectrometer (Picarro G2401-m). It had a laboratory-determined sampling frequency of 1.3–2 Hz and a response time ($t_{10-90\%}$) of 2–3 s. The instrument was calibrated before and after the two research flights using three synthetic gas cylinders from Air Liquide. These secondary standards are traced to the WMO-X2004A scale for methane[47] and the WMO-X2007 scale for $CO_2$[48]. In addition to the Picarro greenhouse gas analyser, the instrumentation package was also comprised of a Licor 7500 for $CO_2$ (20 Hz) and a Licor 7700 for methane (40 Hz). Both yielded similar results to the Picarro, but with higher noise and are therefore not analysed here. A five-hole probe was used to determine the airflow, and merged with position and attitude data to determine the 3D wind vector[49-51].

The morning flight (8:31–11:30 UTC) was originally designed to allow for emission estimation using the classical mass balance technique [e.g. refs. 19,52–57]. It included transfer flight legs to and from the northern leak sites that were, due to flight restrictions, above 600 m and, therefore, above the boundary layer, which was 300–600 m during our measurements (Supplementary Note 2.2), except for a vertical profile close to the southern leak site. At the northern leak sites, we flew a leg upwind of the leak sites inside the boundary layer (mean altitude 116 m), which was intended to determine the atmospheric methane background mole fraction, followed by a number of downwind legs at different altitudes (up to about 200 m, i.e. inside the boundary layer) and distances from the targeted point source, i.e. the leak locations. In addition, the flight included profiles to determine the atmospheric boundary layer height. The minimum distance to the leaks was dictated by aerial exclusion zones, which spanned 5 nautical miles (about 9 km) around the leaks[39-41]. Since we observed the largest methane enhancements during the upwind leg, we adapted the path of the afternoon flight (13:23–16:11 UTC) at short notice to study the methane distribution upwind of the northern leak sites. In total, the focus area around the northern leak sites covered an along-wind corridor 20 km wide and 70 km long, which, with respect to the northern leak sites, extended from 45 km in the upwind direction (southwest) to 25 km in the downwind direction (northeast; Fig. 1). Unlike during the first flight, most of the transfer legs of the second flight could be flown at low altitude (mean 87 m), which allows us to expand the analysis to the south.

We analyse the methane data we obtained inside the boundary layer, as the boundary layer is directly influenced by local sources. Boundary layer observations are selected based on flight altitude and gradients of the virtual potential temperature. The data were averaged over 20-s intervals to reduce the impact of small-scale variability on

covariance parameter estimation and reduce the computational cost, yielding 481 data points.

### Background methane mole fractions

In the top-down (inverse modelling) approach, emissions are estimated from methane enhancements above the background level. Therefore, biases in the background mole fraction lead to biases in the derived emissions. During most of our flight paths, we observed highly variable methane mole fractions, hampering the reliable identification of the methane background. For a few short sections we are confident that they represent background conditions, based on low standard deviation and mean value. They may not be representative for the background throughout the flights, since spatial and temporal gradients could be masked by the local source. Therefore, we use additional data to derive two estimates for the methane background.

**BG CAMS.** For the first background estimate, we sample global inversion-optimised methane mole fraction fields from the Copernicus Global Atmosphere Monitoring Service (CAMS)[58]. The CAMS dataset has a spatial resolution of $2° \times 3°$ and a temporal resolution of 6 h. Since this resolution is coarse compared to our domain, we do not expect a perfect fit to our data. Therefore, we fit an offset to bring the sampled CAMS mole fractions in agreement with the few methane observations in our dataset that most likely represent background values based on low mean and variability. Additional details and results are given in Supplementary Note 1.

**BG $CO_2$.** The second methane background estimate is based on fitting our $CO_2$ observations to the methane data. The $CO_2$ observations exhibited little variability on short scales, but over the course of our two flights, had gradients that correlate with methane in the troughs between peaks (Supplementary Figs. 4, 5). Since we do not expect a substantial local source or sink of $CO_2$, the $CO_2$ gradients likely represent $CO_2$ background variability. We suspect that $CO_2$ and methane backgrounds may have had similar gradients during our observations for two reasons. First, common gradients may be caused by entrainment of air from the free troposphere. The free troposphere had lower $CO_2$ and methane mole fractions than the boundary layer, so entrainment would affect $CO_2$ and methane similarly. Second, common gradients may be due to long-range transport of upwind accumulations of both $CO_2$ and methane, e.g. in a shallow nocturnal boundary layer. On the other hand, $CO_2$ and methane have different spatial and temporal distributions of sources and sinks. Therefore, $CO_2$ background variability does not necessarily fully reflect methane background variability. We mitigate this to some extent by scaling the $CO_2$ observations using the above-mentioned CAMS methane fields, as well as inversion-optimised $CO_2$ fields[58] (1. $9° \times 3.75°$, temporal resolution 3 h). Thus, we obtain the proxy $p_{bg}$ (Eq. (1)) and fit a slope and offset to obtain BG $CO_2$ (Eq. (2)).

$$p_{bg}(t) = CO_2^{obs}(t) \cdot \frac{CH_4^{CAMS}(t)}{CO_2^{CAMS}(t)}, \qquad (1)$$

$$c_{BG\,CO_2}(t) = a \cdot p_{bg}(t) + b. \qquad (2)$$

Additional details and results are given in Supplementary Note 1.

### Atmospheric transport

We link atmospheric observations to surface emissions using the atmospheric Lagrangian particle dispersion model (LPDM) STILT[59] (stochastic time-inverted Lagrangian transport model). For a given location, STILT releases an ensemble of particles backwards in time and computes the time spent in the lower half of the boundary layer. This yields the corresponding surface influence (sensitivity) function

or footprint. The footprint quantifies by how much the atmospheric mole fraction at a given location increases due to a unit emission from the surface.

Footprints are computed along the flight path with a frequency of one second and at the grid spacing of the emission estimation grid (Section "Emission estimation method"). For emission estimation, they are averaged over the same 20-s intervals as the observations.

We drive STILT with the WRF (Weather Research and Forecast) model, version 3.9.1.1[60]. The WRF domain covers most of Europe at a horizontal grid spacing of 10 km, and a nest with a grid spacing of 2 km covers the southern part of the Baltic Sea. The runs have 81 vertical layers, 3–5 of which are inside the PBL during our observations (the lowest layer boundaries are at 0, 60, 140, 240, 380, 540, 760 and 1010 m; rounded to 10 m). WRF is driven by ERA5, the fifth-generation ECMWF atmospheric reanalysis of the global climate and weather[61–63].

### Assessment of transport uncertainty

A major uncertainty in the top-down (inverse modelling) approach to emission quantification is transport model uncertainty, but it is often not quantified. We provide a limited assessment of transport uncertainty by two different approaches. First, we slightly vary the WRF setup (Meteo A and Meteo B), yielding two sets of meteorological fields for STILT (Section "Meteorological fields"). Second, we conduct a simple experiment comparing Eulerian vs. Lagrangian transport (Section "Eulerian vs. Lagrangian transport"). All setup variations perform similarly well with respect to our meteorological observations (Supplementary Note 2); hence, we do not regard any one setup as being the most likely.

**Meteorological fields.** We derive meteorological fields from two WRF runs, dubbed Meteo A and Meteo B. The runs use the same domain, physics and dynamics options. To ensure consistency with weather observations, both WRF runs use grid nudging towards ERA5 in the outer domain. Grid nudging means that non-physical tendency terms are applied in the meteorological simulation that keeps the deviation from ERA5 small. In Meteo A, the inner domain is also nudged towards ERA5, whereas in Meteo B, the inner domain is nudged towards the wind measurements obtained during our flights instead. Meteo A was originally designed to simulate transport from the beginning of the Nord Stream event, and thus started on 25 September 18:00 UTC. Since our observations only cover 5 October, Meteo B was started much later, on 4 October 18:00 UTC, to avoid drift of the simulated meteorological fields in the inner domain away from observed meteorological conditions.

**Eulerian vs. Lagrangian transport.** There are two fundamental ways to simulate atmospheric transport, the Eulerian and the Lagrangian approach. Lagrangian models, such as the STILT model that we use to optimise emissions, trace the locations of particles as they are transported by meteorological fields from a numerical weather prediction model (here: WRF). Eulerian models compute changes in concentrations on a discrete grid in space and time.

We quantify the impact of the transport algorithm using WRF-GHG, a WRF module that simulates atmospheric greenhouse gas

transport using the Eulerian approach[64]. Specifically, we run WRF-GHG with the surface emissions estimated using STILT and the same settings used for computing STILT footprints. Then we sample the resulting methane fields at the measurement locations and compute a single scaling factor for the emissions such that the sampled simulated methane mole fractions best fit the observations. The difference between unscaled and scaled emissions allows us to quantify the differences between the Eulerian and Lagrangian approaches.

### Emission estimation method

To estimate emissions from our atmospheric observations, we use the inverse model of atmospheric methane transport of ref. 65, available at https://github.com/greenhousegaslab/geostatistical_inverse_modeling[66]. Here, we introduce the method briefly. More details are given in Supplementary Methods 2.

The basic transport equation is

$$\Delta\mathbf{c} = \mathbf{c_{obs}} - \mathbf{c_{bg}} = \mathbf{H} \cdot \mathbf{f} + \boldsymbol{\epsilon}. \tag{3}$$

Here, $\Delta\mathbf{c}$ is a vector of atmospheric methane mole fraction enhancements (observed $\mathbf{c_{obs}}$ minus background $\mathbf{c_{bg}}$), $\mathbf{H}$ is the transport matrix (composed of the footprints described in the section "Atmospheric transport"), $\mathbf{f}$ is a vector of the fluxes to solve for, and $\boldsymbol{\epsilon}$ are errors. Since solving Eq. (3) for $\mathbf{f}$ is generally an underconstrained problem, we regularise it by minimising model-data mismatch while requiring that the solution stay close to a prior flux estimate. Ideally, an inverse model features an unbiased prior flux estimate. However, the spatial distribution of the fluxes from the ocean model is at odds with the atmospheric data (Fig. 3d). Therefore, we provide an independent emission estimate by setting the mean prior flux estimate to zero throughout the domain instead of using the ocean model results as prior flux estimate (note that despite the bias in the prior flux towards zero, atmospheric enhancements modelled using the posterior flux are unbiased; Table 1). Thus, we find the flux estimate $\mathbf{f_{post}}$ that minimises the cost function $J(\mathbf{f})$:

$$J(\mathbf{f}) = (\Delta\mathbf{c} - \mathbf{H} \cdot \mathbf{f})^T \mathbf{R}^{-1} (\Delta\mathbf{c} - \mathbf{H} \cdot \mathbf{f}) + \mathbf{f}^T \mathbf{Q}^{-1} \mathbf{f}, \tag{4}$$

with model-data mismatch covariance matrix $\mathbf{R}$ and prior flux covariance matrix $\mathbf{Q}$. We modify the solution $\mathbf{f_{post}}$ by restricting it to non-negative values using a Lagrange multiplier method[67].

For simplicity, the model-data mismatch matrix $\mathbf{R}$ in the Bayesian cost function (Eq. 4) is chosen to be diagonal with variance $r^2$ for all observations. The prior flux covariance matrix $\mathbf{Q}$ is constructed from prior flux variances $q^2$ and a spatial correlation length $d$. The correlation length often refers to a Gaussian correlation model. For computational efficiency, we use a spherical covariance model instead (Supplementary Equation 3). In this model, correlations at distances greater than $d$ are zero. We estimate the covariance parameters $r$, $q$ and $d$ using a maximum likelihood method[68].

The domain for emission estimation covers the area 14.9–16.2°E, 54.6–55.8°N at a grid spacing of $(1/120)°$ in the latitudinal direction and $(1/80)°$ in the longitudinal direction, which yields grid cells with an area of ~0.7 km². We do not resolve potential temporal variations in

**Table 2 | Short description of key aspects of the ocean model**

| | |
|---|---|
| Dissolved methane at leak locations | Results of the near-field model from ref. 20 |
| Total methane dissolved [kt] | 10.8 |
| Ocean currents | Baltic Sea Physics Analysis and Forecast[43], resolution: 1' latitude, 1'40" longitude, 55 vertical levels |
| Transport of dissolved methane | Lagrangian, horizontal advection and diffusion in the mixed layer. Diffusion calibrated using observations of dissolved methane from DE-SOOP Finnmaid[70] |
| Outgassing | Wanninkhof[71], driven by ERA5 winds |

Additional details are given in Supplementary Methods 3.1.

the emissions, since our measurement coverage is better suited to provide a mean estimate.

## Ocean model

The inputs and components of the ocean model are (a) methane release from the pipelines, (b) dissolution from the bubbles into the plume water at the leak locations, (c) transport of dissolved methane in the water and (d) outgassing to the atmosphere. A detailed model description is given in Supplementary Methods 3.1. Important aspects are summarised in Table 2.

## Data availability

The atmospheric dataset is available at https://doi.org/10.18160/D0DQ-F7GE[69]. DE-SOOP Finnmaid data are available at https://doi.org/10.18160/K3BM-8YNG[70].

## Code availability

The current version of the inverse model is available at https://github.com/greenhousegaslab/geostatistical_inverse_modeling[66]. Ocean model code and custom code used for the analysis of the data are available from the authors upon reasonable request.

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

## Acknowledgements

We would like to thank Technisches Hilfswerk Landesverband Bremen and Niedersachsen for logistical support of the flight campaign. Thanks to Bartłomiej Kieblesz (Helipoland), who piloted the

helicopter. Thanks to Christoph Gerbig (MPI-BGC) for helpful discussions on the STILT model and Bastien Queste (Uni Gothenburg) for sharing insights on Baltic Sea circulation. We thank Michael Glockzin (IOW) for their work on the SOOP Finnmaid data. Thanks to Vineet Yadav (JPL) for providing the inverse modelling code that we used to compute posterior covariance matrices. This study has been conducted using E.U. Copernicus Marine Service Information; doi:10.48670/MOI-00010. This study used resources of the Deutsches Klimarechenzentrum (DKRZ) granted by its Scientific Steering Committee (WLA) under project ID bd1231. The German Federal Ministries for Education and Research (BMBF) and Digital and Transport (BMDV) are acknowledged for support of ICOS Germany. This work has been supported by funding from UNEP's International Methane Emissions Observatory (IMEO).

## Author contributions

A.R., J.M., and F.R. conceptualised the study. A.R. and H.H. organised the observation campaign. L.B., K.G., A.L., M.L., J.M., F.P., M.P., F.R., and A.R. planned and conducted the observation flights. F.P., M.L., and M.P. calibrated and controlled the quality of the atmospheric data. H.B. and G.R. provided the DE-SOOP Finnmaid data. H.B. did the time-lapse correction of the DE-SOOP Finnmaid data. F.R. and J.M. performed atmospheric transport simulations. T.G. analysed the meteorological data and simulations. F.R. and S.M. performed emission estimation. A.D., J.M., and M.M. performed ocean process simulations. F.R., G.B., A.D., J.G., M.M., and G.R. shaped the interpretation of the ocean model results. F.R. wrote the manuscript with input from all authors.

## Funding

## Competing interests

The authors declare no competing interests.
