## [Peer Review File · Nature Communications]

Airborne observations reveal the fate of the methane from the Nord Stream pipelinesEditorial Note: This manuscript has been previously reviewed at another journal that is not operating a transparent peer review scheme. This document only contains reviewer comments and rebuttal letters for versions considered at *Nature Communications*.

REVIEWERS' COMMENTS

Reviewer #1 (Remarks to the Author):

This paper reports estimates of methane emissions a few days after the Nord Stream pipeline leak that occurred on 26th September 2022. Measurements taken from an airborne platform on 5th October are combined with inverse modelling techniques to derive an estimate of the emissions occurring from methane dissolved in the ocean following the initial leak. These are compared with the results of an ocean model and reveal uncertainties in the model, especially the spatial distribution.

The work is novel by its nature as events like the Nord Stream pipeline leak do not occur very frequently. It provides novel insights into the fate of methane after a large undersea leak and demonstrates the importance of in-situ airborne observations to quantify ethane emissions, with the amount being released from the ocean being of similar magnitudes to the largest onshore anthropogenic sources.

The data analysis methods are robust and very well described in the manuscript (including the SI) and the work will be important in analysing data from potential future similar events.

I recommend publication upon dealing with the following minor comments.

The authors could comment more in the main text about the errors in the BG CO₂ methodology. CO₂ and CH₄ are clearly from different sources and with different lifetimes so it is important to understand the potential flaws in this methodology.

How does the modelled BL compare with measurements from profiles? Does this affect the error (line 447). It is in the SI but might be worth stating in the main text.

In the conclusions, the authors say something about the length of time this outgassing from the ocean is expected to last. This can lead on to a discussion about how these emissions compare to the initial short-lived emission immediately following the explosions and put the impact of the leak on a global scale.

Minor:

In the introduction:

Line 50: Used anthropogenic twice in this sentence, please reword.

Line 55: What percentage of methane is estimated to come from the natural gas supply chain?

Table 1: What measure of correlation is used?

Reviewer #2 (Remarks to the Author):

Comments on Reum et al. : Airborne Observations – Nord Stream pipelines

General Comments

This is a careful, detailed account of fast-response measurements after the giant Nord Stream methane blow out on 26 Sept. 2022. It was an important event, releasing as much methane as the annual emissions of many nations, and urgently needed quantification. The authors are to be congratulated on getting into the air by 5 Oct.

The evidence-gathering was intelligent, the paper is detailed and carefully thought through, and the conclusions appear valid. I have some suggestions but they are only in the hope of adding to the work's usefulness. The main recommendation is to ask for a paragraph or two outlining suggested improvements to the response planning, in case a similar event happens in future, somewhere else.

The paper is important and should be published with minor revisions.

Specific points

Abstract: - line 040. the total emissions from the event should also be mentioned (I appreciate that number comes from a parallel paper, but each paper should have stand-alone usefulness. The number is given in the main text: line 078.

Line 084 Left – for comparison, give total and hourly emission estimates from these events, especially Aliso.

Line 074-092 right – mention methanotrophy is slow and maybe give rates? (I see later on Line 298R this is mentioned but might be better to bring it upfront here).

Line 150 left – get a long cable for the towed instrument? The Elgin gas leak study had similar ‘blow up the aircraft’ fears! Indeed, maybe mention Lee, James D., et al. (2018) Flow rate and source reservoir identification from airborne chemical sampling of the uncontrolled Elgin platform gas release. Atmospheric Measurement Techniques 11: 1725-1739.

Line 133 right – ‘both in the west’ – I was puzzled here. Both whats? Does ‘both’ refer to west and south? Maybe rewrite sentence a little.

L176 Fig 1. Maybe mention wind direction in the caption – it took me a while to find the wind arrow. Incidentally, normal military procedure is to intercept at 30 degrees – that gives a much longer intercept. Going across at 90 degrees can mean the instrument is only just catching the plume before you are out of it. (see comment later for ‘future advice’).

L190 right – isolated hot spot – interesting. Any more detail available?

L237 left – English error – “the sum.....are”

L254 left – any reference for the ocean circulation pattern? Anyway, the Baltic is barely an ‘ocean’!

L239 right – outgassing model...details?

L300 right – 30 t /hr – uncertainty/error in this number?

L324 left – 0.3 correlation...interesting puzzle. See also Line 321 right and L342 and L352.

Conclusions.

It would add greatly to the paper if there was a ‘future’ paragraph – a recipe list, suggesting how to

prepare and what to do if something like this happens again – and it will (Aliso, Elgin, etc etc). I am especially sad that no $\delta^{13}\text{C}(\text{CH}_4)$ isotopic measurements were made as these would a) have helped assess the proportionate impact methanotrophy and dissolution, and b) directly proven it was Nordstream gas being measured (e.g. see Cain, Michelle, et al. (2017) A cautionary tale: A study of a methane enhancement over the North Sea. JGR: Atm 122: 7630-7645.) OK in this case it's pretty obvious where the gas came from, but maybe not around a smaller pipe in a shallow sea near wetlands or cows. I'd suggest the paper should recommend at least air sampling in flasks or bags, even if an isotopic instrument can't be flown on board.
Methodology - appropriate.

Overall, a good paper that should be published after minor changes.

Reviewer #2 (Remarks on code availability):

N/A

Reviewer #3 (Remarks to the Author):

Review of Reum et al. : Airborne observations reveal the fate of the methane from the Nord Stream pipelines

The manuscript presents interesting results on measured methane fluxes right after the leaks from the Nord Stream pipelines had ceased. The amount and extend of the leaks have been studied in several papers duly cited in the manuscript. Nevertheless, the manuscript brings new information on the subject and also examines sources of uncertainties in the estimations. The manuscript is one of a series of three manuscripts handling the leaks from different perspective and are partly interconnected. The focus of the manuscript is on the air-borne measurements of the methane fluxes, while the oceanic part is mostly used as supporting information. The analysis of the ocean model/models results is interesting in regard of the performance of the models, but a more detailed study would require a study of its own.

Although reference [36] and especially reference [28] was not available, the manuscript contained enough information to evaluate it as an independent manuscript and to judge the conclusions made. I have some comments on the manuscript that I would like the authors to respond.

Comments:

Chapter 1

Page 2, second column, line 067: what kind of wind conditions there were prior to the measurements? I suppose more information would be in reference [28] which is not yet published. Mixing of the surface layer is dependent on the wind, wave and thermohaline conditions which may be different from those presented in [29]. See also the comment on Chapter S1.2.1.

Page 2, second column, line 089: the depth of the mixed layer: I could not find this information in chapter "Methods". In addition, please add relevant chapter number when referring to the chapter "Methods" throughout the manuscript.

Chapter 2.1

Page 3, first column, line 136: it might be good to define "upwind" at this point. Using upwind and downwind instead of SW and SE is a bit vague since the wind conditions and circulation in the upper ocean before the flight are not studied. Nevertheless, the wind direction is clearly shown in the Figures so the choice between wind direction and geographical coordinates is not critical.

Page 3, second column, line 118 and Fig. 1: it seems more like the extent in the downwind direction was cut in the middle of the outgassing area. Too bad that the flight time did not allow for a longer track.

Chapter 2.2.1

Page 4, first column, lines 191, 198 and Figure 2: define what is meant by "sensitive"/"aggregated sensitivity". They are not explained in Chapter 4 or Supplementary material.

Page 4, first column, lines 193-194: the lowest values are from the combination Meteo B and BG CO₂. The modelled wind agreed with measurements (see also Chapter S2.3): does the difference stem from uncertainties with CO₂? See also comment on Chapter 4.2 below.

Chapter 2.3

Page 5, second column, line 231: since the Baltic Sea does not have permanent currents but are mainly driven by winds, it might be better to examine what the circulation was during the days of the leaks rather than looking at the long-term mean circulation to confirm that the gyre existed during this time.

Page 7, first column, lines 338->: see the comment above. On lines 346-347 it is stated that that the results in [28] show that in longer timescales the gyre is visible. Was the gyre not clearly visible after 6 days and if so, is there a reason to suspect that the model results were not reliable? What was the resolution of the ocean model? Can the grid resolution explain the lack of detail in the spatial distribution?

Page 7, second column, lines 338-> : The exact form of the transfer velocity is not yet known, so there are several formulations for it. What was the reason for using a transfer velocity formulation that differs from the one used in M23? According to Figures S7 and S13, the transfer velocity is nearly twice as high for the wind speeds observed during the flight. If the only difference would be the vertical distribution, its role could be better evaluated.

Page 8, first column, line 374. Could you explain why overestimation of transfer velocity would cause smaller emissions?

Chapter 4.2

Page 10. first column, line 509. Referring to the Supplementary material chapter S2.1 and Figure S4: what might have been the reason for the large differences between the CAMS-CO2 and measured CO2 during flight 2 at low altitudes? See also the second comment on Chapter 2.2.1 above.

Chapter 4.3

Page 11, first column, lines 511-514: when defining the footprint of the observed fluxes, was the atmospheric stratification taken into account or was it assumed to be neutral? The stratification has a strong impact on the location from where the flux originates. This can be important since the ocean model gave larger spatial distribution.

Page 11, first column, lines 554, 557: what do words "nudge towards" mean? A more precise formulation would be helpful.

Page 11. second column, line 511: what drift needed to be avoided?

Chapter 4.4

Page 11, second column, lines 553-560: is it realistic to assume that the mean prior flux estimate is zero? How much would the mentioned use of ocean model results (line 560) change the situation?

Chapter S1.2.1

Since the spatial distribution given by the ocean model is blurred compared to the atmospheric measured, could this indicate that the assumption of evenly distributed particles is not correct? The rising gas bubbles caused mixing and turbulence. The modelled mixed layer depth was at a depth of 20-25 m (Figure S11): how reasonable is the assumption that the methane was mixed instantly to the whole layer? That is, were the conditions such that the mixing layer was mixed at least to the depth of the mixed layer in the given time frame? Was the radius of the circle describing the gas plume constant in the mixed layer in the model used?

Minor issues

According to the reference list, references [11] and [28] will be/are submitted to the same journal. It might be good to mention this in the outline of the manuscript in Chapter 1.

Chapter 4.2

Page 10, first column, line 482: above a background level?

Reviewer #3 (Remarks on code availability):

The work consists of multiple parts and rather heavy calculations: the codes should be run on a supercomputer which was not possible for this reviewer. The readme-files were carefully written and at least superficially judged usable for potential user. Also the arrangement of the material was clear. The reference to Wanninkhof 2014 was missing from the main readme-file under Ocean model.

Response to reviews

Manuscript: “Airborne observations reveal the fate of the methane from the Nord Stream pipelines”

Authors: Friedemann Reum, Julia Marshall, Henry C. Bittig, Lutz Bretschneider, Göran Broström, Anusha L. Dissanayake, Theo Glauch, Klaus-Dirk Gottschaldt, Jonas Gros, Heidi Huntrieser, Astrid Lampert, Michael Lichtenstern, Scot M. Miller, Martin Mohrmann, Falk Pätzold, Magdalena Pühl, Gregor Rehder, Anke Roiger

Reviewer comments are shown in black, responses in blue.

Reviewer #1 (Remarks to the Author):

This paper reports estimates of methane emissions a few days after the Nord Stream pipeline leak that occurred on 26th September 2022. Measurements taken from an airborne platform on 5th October are combined with inverse modelling techniques to derive an estimate of the emissions occurring from methane dissolved in the ocean following the initial leak. These are compared with the results of an ocean model and reveal uncertainties in the model, especially the spatial distribution. The work is novel by its nature as events like the Nord Stream pipeline leak do not occur very frequently. It provides novel insights into the fate of methane after a large undersea leak and demonstrates the importance of in-situ airborne observations to quantify ethane emissions, with the amount being released from the ocean being of similar magnitudes to the largest onshore anthropogenic sources. The data analysis methods are robust and very well described in the manuscript (including the SI) and the work will be important in analysing data from potential future similar events. I recommend publication upon dealing with the following minor comments.

The authors could comment more in the main text about the errors in the BG CO₂ methodology. CO₂ and CH₄ are clearly from different sources and with different lifetimes so it is important to understand the potential flaws in this methodology.

This information is given in the Methods section on “BG CO₂”. To make it easier to find, we now summarize it in the section “Methane emissions estimated from atmospheric data” of the main text. Note, we prefer the terms “different sources and sinks” instead of “different sources and different lifetimes”:

The ambiguity in the methane background (Sect. 2.1) requires relying on additional data sources and assumptions. In particular, the upper bound of the methane background, which determines the lower bound of the emission estimates, relies on the observation that CO₂ and methane backgrounds could have been strongly correlated during our measurement campaign (e.g. due to entrainment from the free troposphere or long-range transport of accumulations). This assumption is uncertain since CO₂ and methane have different sources and sinks, and therefore, we use it only to derive a lower bound of emissions (see Methods for details on the methane background estimation).

How does the modelled BL compare with measurements from profiles? Does this affect the error (line 447). It is in the SI but might be worth stating in the main text.

We now summarize the findings stated in the SI on the profiles in the results section “Methane emissions estimated from atmospheric data”:

Before:

Differences in the modelled boundary layer heights contribute to the uncertainty (Supplementary results S2.4).

Revised:

There is a mean difference in the boundary layer heights between the two meteorological simulations. However, in most cases, both simulations agree with pronounced capping inversions that we observed in the morning (flight 1), making it difficult to differentiate the accuracy of the boundary layer heights of the two simulations despite their mean difference (Supplementary Note 2.2).

In the conclusions, the authors say something about the length of time this outgassing from the ocean is expected to last. This can lead on to a discussion about how these emissions compare to the initial short-lived emission immediately following the explosions and put the impact of the leak on a global scale.

We give the estimated total emissions and a comparison to put them into perspective in the introduction (citing the companion manuscript Harris et al.). However, we agree that it’s useful to mention the amount of methane that dissolved, to put it in relation to the total emissions released during the event. Therefore, we add to the introduction:

The total amount of methane that dissolved was 9-15 kt [3].

Minor:

In the introduction:

Line 50: Used anthropogenic twice in this sentence, please reword.

We edited the text accordingly.

Line 55: What percentage of methane is estimated to come from the natural gas supply chain?

We replace the general statement that emissions related to natural gas are „important“ with the following more precise statement:

Inventory data put recent emissions from the largest natural gas supply chains at 26 Tg CH₄ yr⁻¹ globally[4] and total emissions related to oil and gas exploitation at 80 Tg CH₄ yr⁻¹, representing 63 % of anthropogenic methane emissions. These estimates have large uncertainties [2, 4]. Atmospheric observations have revealed that emissions in inventories can be underestimated because few hot spots can dominate regional budgets [e.g., 5].

Table 1: What measure of correlation is used?

We add that information to the caption of the Table:

The correlations are Pearson correlation coefficients.

We would like to thank Reviewer #1 for their constructive comments!

Reviewer #2 (Remarks to the Author):

Comments on Reum et al. : Airborne Observations – Nord Stream pipelines

General Comments

This is a careful, detailed account of fast-response measurements after the giant Nord Stream methane blow out on 26 Sept. 2022. It was an important event, releasing as much methane as the annual emissions of many nations, and urgently needed quantification. The authors are to be congratulated on getting into the air by 5 Oct.

The evidence-gathering was intelligent, the paper is detailed and carefully thought through, and the conclusions appear valid. I have some suggestions but they are only in the hope of adding to the work's usefulness. The main recommendation is to ask for a paragraph or two outlining suggested improvements to the response planning, in case a similar event happens in future, somewhere else.

The paper is important and should be published with minor revisions.

Specific points

Abstract: - line 040. the total emissions from the event should also be mentioned (I appreciate that number comes from a parallel paper, but each paper should have stand-alone usefulness. The number is given in the main text: line 078.

We add the total emissions from Harris et al. to the first sentence (note that the estimated was updated during the revisions of Harris et al. from 445 ± 25 kt to 465 ± 20 kt):

The Nord Stream pipeline leaks on 26 September 2022 ~~led to~~ released 465 ± 20 kt of methane into the atmosphere, the largest recorded transient anthropogenic methane emission event. While most of the ~~methane gas~~ escaped directly to the atmosphere, ~~but~~ a fraction dissolved in the water

Line 084 Left – for comparison, give total and hourly emission estimates from these events, especially Aliso.

We included the estimates:

It far exceeds other exceptional transient anthropogenic methane emission events that have been quantified using atmospheric observations from aircraft and space, such as well blowouts in Aliso Canyon (up to 60 t h⁻¹ and 97.1 kt total) [15], Louisiana (up to 101 (49 – 127) t h⁻¹ and 49 (21 – 63) kt total) [16], Ohio (120 ± 32 t h⁻¹ and 60 ± 15 kt total) [17], Texas (up to 27.6 ± 8.8 t h⁻¹ and 4.8 ± 1 kt total) [18] and the Elgin platform release (up to 4.7 ± 0.7 t h⁻¹) [19].

Line 074-092 right – mention methanotrophy is slow and maybe give rates? (I see later on Line 298R this is mentioned but might be better to bring it upfront here).

We prefer to retain the timescales we gave in this paragraph to compare methanotrophy (months) and outgassing (days) instead of rates. However, to emphasize the point more clearly that methanotrophy was likely much slower than outgassing, we add the following statement here:

Methane is removed from sea water by outgassing to the atmosphere [31] and microbial oxidation. In the first days after the pipeline explosions, outgassing likely dominated because microbial methane oxidation typically takes months in the Baltic Sea [32, 33], and the timescale for methanotrophic microbial communities to adapt to an increased methane availability is on the same order [34].

Line 150left – get a long cable for the towed instrument? The Elgin gas leak study had similar ‘blow up the aircraft’ fears! Indeed, maybe mention Lee, James D., et al. (2018) Flow rate and source reservoir identification from airborne chemical sampling of the uncontrolled Elgin platform gas release. Atmospheric Measurement Techniques 11: 1725-1739.

In principle, a considerably longer rope may alleviate aircraft safety concerns in some cases. However, a longer rope would also involve challenges during the operation of the sling load. Therefore, we do not plan on using a longer rope with the HELiPOD at present. In the Nord Stream case, the flight restriction zones were in place for up to 3500 ft (~1 km; we made the description of the flight restriction zones in the revised manuscript more precise and included this altitude), at least 400 m above the boundary layer. We were in contact with the authorities about the flight restrictions prior and during our flights. While we were successful in obtaining the permission to do the transfer legs of flight 2 inside the boundary layer (which we were not allowed during flight 1), we could not obtain a permission to enter the flight exclusion zones close to the leaks. We believe that a higher flight altitude with a longer cable would not have increased our chances, as the flight exclusion zones were probably not only related to aircraft safety, but also to security (the leak sites were essentially a crime scene under investigation at the time).

For the sake of brevity, we do not include these considerations in the manuscript. As suggested by the reviewer, we add here that Lee et al. (2018) faced similar flight restrictions:

These flight exclusion zones were similar to the one that had been in place during the flights that were undertaken to quantify methane emissions from the Elgin platform release in 2018 [19].

Line 133 right – ‘both in the west’ – I was puzzled here. Both whats? Does ‘both’ refer to west and south? Maybe rewrite sentence a little.

Yes, the “both” referred to “west” and “south”. We expanded the sentence to make it clearer:

However, the lowest methane concentrations at the northern edge were up to 50 ppb higher than the lowest ones during the campaign, which were observed in the west (i.e. upwind) of the northern leaks and south of the southern leak.

L176 Fig 1. Maybe mention wind direction in the caption – it took me a while to find the wind arrow. Incidentally, normal military procedure is to intercept at 30 degrees – that gives a much longer intercept. Going across at 90 degrees can mean the instrument is only just catching the plume before you are out of it. (see comment later for ‘future advice’).

Done:

Paths of the two flights in grey, with leak locations as black triangles, flight exclusion zones as orange circles, and the black arrow indicating the average wind direction (217°). ...

L190 right – isolated hot spot – interesting. Any more detail available?

We did not investigate the reason behind that feature. However, Abrahamsson et al. recently published a study (<https://doi.org/10.1038/s41598-024-63449-2>) that shows a similar feature. We add a reference to that in our text:

Before:

It may be related to the larger emission band but disconnected due to spatial data sparsity.

Revised:

It could be related to the larger emission band but disconnected due to spatial data sparsity. Note that ship borne measurements revealed a hot spot of dissolved methane close-by in the period 3-6 October 2022 [37].

L237 left – English error – “the sum.....are”

Thanks! Changed to “*the sum... is*”

L254 left – any reference for the ocean circulation patter? Anyway, the Baltic is barely an ‘ocean’!

As reference, we had cited Placke et al. 2018 (<https://doi.org/10.3389/fmars.2018.00287>; ref. 38 in the initial submission) about the Bornholm Basin gyre. However, to show the conditions that affected the dissolved methane from the Nord Stream leaks until our measurements – also in response by a comment by Reviewer #3 – we added a plot of the mean surface currents from 26 September to 5 October, derived from the ocean circulation product that we used in our model (CMEMS, then ref. 60). The results are shown in the new Supplementary Fig. 14.

We agree that the Baltic Sea is not an ocean, but decided to retain the selected wording since the general concepts are applicable to other seas and oceans, similarly like the above-mentioned paper by Placke et al. used the term “ocean circulation model” in their work about the circulation in the Baltic Sea.

L239 right – outgassing model...details?

The outgassing component of the model is an implementation of Wanninkhof 2014. To avoid confusion, we add a cross-reference to the method description here:

... coupled to a simple Lagrangian tracer dispersion and outgassing model developed for this study (see Methods and Supplementary Methods 3.1).

L300 right – 30 t/hr – uncertainty/error in this number?

We do not provide an uncertainty estimate for this number, as the central estimate is sufficient for the main purpose of the ocean model – to confirm that the methane we observed originated from the Nord

Stream leaks. A thorough uncertainty assessment would involve examining the potential uncertainties in the ocean models we identified in the discussion, including uncertainties “in the total amount of dissolved methane, in the vertical distribution of dissolved methane and the gas transfer velocity”. Such an investigation is out of the scope of our paper, which focuses on what we can learn from our atmospheric dataset. Note that, later in the manuscript, we present the results of the similar ocean model by Mohrmann et al., who provide an uncertainty estimate that yields a range of 13 (11-18) t h⁻¹ on 5 October 2022. The range is dominated by the uncertainty of the amount of initially dissolved methane. We hope that our findings stimulate further investigation by the research community of the uncertainty of the emission estimates from the ocean models.

L324 left – 0.3 correlation...interesting puzzle. See also Line 321 right and L342 and L352.

Indeed. We hope that our interpretation, which is given in the subsequent lines, is clear.

Conclusions.

It would add greatly to the paper if there was a ‘future’ paragraph – a recipe list, suggesting how to prepare and what to do if something like this happens again – and it will (Aliso, Elgin, etc etc). I am especially sad that no d13C(CH₄) isotopic measurements were made as these would a) have helped assess the proportionate impact methanotrophy and dissolution, and b) directly proven it was Nordstream gas being measured (e.g. see Cain, Michelle, et al. (2017) A cautionary tale: A study of a methane enhancement over the North Sea. JGR: Atm 122: 7630-7645.) OK in this case it’s pretty obvious where the gas came from, but maybe not around a smaller pipe in a shallow sea near wetlands or cows. I’d suggest the paper should recommend at least air sampling in flasks or bags, even if an isotopic instrument can’t be flown on board.

Those are good points, and we’re happy to add the following paragraph to the discussion:

Typically, planning international airborne atmospheric measurement campaigns starts months or even years in advance. In the accelerated timeline for this mission - being airborne nine days after the explosions - we committed personnel and funding, prepared and integrated the science payload with the platform, organized logistics (e.g. transport), hired aircraft and pilot, obtained flight permissions and planned the flights. Potential future fast response missions could be improved in several ways. Additional insights on the origin and fate of leaked gas could be gained by measuring the isotopic signature of methane and the mole fractions of other hydrocarbon compounds [19, 45, 46]. We performed a second airborne campaign to the Nord Stream leak sites in November 2022 where we measured $\Delta^{13}\text{C}-\text{CH}_4$ online and in air samples. Analysis of these data is pending. A more fine-grained characterization of the boundary layer height could also have been beneficial. However, the best allocation of limited available flight time has to be decided on a case-by-case basis and balance spatial sampling density, repetition and spatial coverage.

Methodology - appropriate.

Overall, a good paper that should be published after minor changes.

Reviewer #2 (Remarks on code availability):

N/A

We would like to thank Reviewer #2 for their constructive comments!

Reviewer #3 (Remarks to the Author):

Review of Reum et al. : Airborne observations reveal the fate of the methane from the Nord Stream pipelines

The manuscript presents interesting results on measured methane fluxes right after the leaks from the Nord Stream pipelines had ceased. The amount and extend of the leaks have been studied in several papers duly cited in the manuscript. Nevertheless, the manuscript brings new information on the subject and also examines sources of uncertainties in the estimations. The manuscript is one of a series of three manuscripts handling the leaks from different perspective and are partly interconnected. The focus of the manuscript is on the air-borne measurements of the methane fluxes, while the oceanic part is mostly used as supporting information. The analysis of the ocean model/models results is interesting in regard of the performance of the models, but a more detailed study would require a study of its own.

Although reference [36] and especially reference [28] was not available, the manuscript contained enough information to evaluate it as an independent manuscript and to judge the conclusions made. I have some comments on the manuscript that I would like the authors to respond.

Comments:

Chapter 1

Page 2, second column, line 067: what kind of wind conditions there were prior to the measurements? I suppose more information would be in reference [28] which is not yet published. Mixing of the surface layer is dependent on the wind, wave and thermohaline conditions which may be different from those presented in [29]. See also the comment on Chapter S1.2.1.

The most important piece of evidence that justifies the assumption of fast mixing of the surface mixed layer in the Nord Stream case is that Mohrmann et al. (ref. [28] in the original submission) observed a well-mixed layer indicated by the absence of persistent vertical methane gradients. Therefore, we decided to corroborate the point of a well-mixed layer by referring to Mohrmann et al., instead of using the model from ref. [29] in the revised manuscript:

In the Nord Stream case, vertical mixing was fast in the mixed layer due to wind-induced turbulence, as indicated by homogeneous concentrations of dissolved methane in the mixed layer without consistent depth gradients [30].

To answer the specific question, here are the 10 m winds at the Northern leak sites, obtained from our WRF run A:

Fig. 1: 10-m winds at the northern leak location modeled using our WRF run A.

Except for a ~48 hour period on 28-29 September with calmer conditions, wind speeds were 5-12 m s^{-1} . The median over the entire period is 8.7 m s^{-1} . For the sake of brevity, we do not include the figure in the manuscript.

Page 2, second column, line 089: the depth of the mixed layer: I could not find this information in chapter "Methods".

Apologies, it was the wrong cross-reference. We correct it from “see Methods” to “see Supplementary Figures 11 and 12”.

In addition, please add relevant chapter number when referring to the chapter "Methods" throughout the manuscript.

It looks to us like we should keep referring only to “Methods” instead of chapter numbers, as this seems to be the style of other published papers in Nature Communications. Recent examples: <https://www.nature.com/articles/s41467-024-47569-x>
<https://www.nature.com/articles/s41467-024-46920-6>

Chapter 2.1

Page 3, first column, line 136: it might be good to define "upwind" at this point. Using upwind and downwind instead of SW and SE is a bit vague since the wind conditions and circulation in the upper ocean before the flight are not studied. Nevertheless, the wind direction is clearly shown in the Figures so the choice between wind direction and geographical coordinates is not critical.

We now indicate the direction explicitly here: “upwind (southwest)” and “downwind (northeast)”.

Page 3, second column, line 118 and Fig. 1: it seems more like the extent in the downwind direction was cut in the middle of the outgassing area. Too bad that the flight time did not allow for a longer track.

Yes, we might not have “seen” emissions especially east of the transfer legs. This point is discussed as one potential error source in the manuscript.

Chapter 2.2.1

Page 4, first column, lines 191, 198 and Figure 2: define what is meant by "sensitive"/"aggregated sensitivity". They are not explained in Chapter 4 or Supplementary material.

To clarify the term “sensitivity”, we

- added that the sensitivity is expressed as footprints in line 191 (the term “footprint” is explained in the Methods section):

Owing to the wind direction, the observations were sensitive to emissions southwest of the flight tracks, as indicated by their footprints and the reduction of flux uncertainty (Fig. 2).

- replaced the term “aggregated sensitivity” in the caption of Fig. 2:

(a) ~~Aggregated sensitivity of observations to emissions and~~ Sum of footprints (surface influence functions) of all observations, indicating the area to which the observations are sensitive, (b)...

Page 4, first column, lines 193-194: the lowest values are from the combination Meteo B and BG CO2. The modelled wind agreed with measurements (see also Chapter S2.3): does the difference stem from uncertainties with CO2? See also comment on Chapter 4.2 below.

The main contributions to the range of emission estimates (boundary layer height and methane background) were given in the discussion. However, this comment and comments by Reviewer #1 indicate to us that we should mention them here already. Therefore, we add to the paragraph:

The uncertainty in the total emissions is due mostly to uncertainty in the boundary layer height and in the methane background mole fractions, which we describe in the following paragraphs.

...

Uncertainty in CO2 did not affect our emission estimates (see also our response to the comment on Chapter 4.2).

Chapter 2.3

Page 5, second column, line 231: since the Baltic Sea does not have permanent currents but are mainly driven by winds, it might be better to examine what the circulation was during the days of the leaks rather than looking at the long-term mean circulation to confirm that the gyre existed during this time.

We added a plot to the supplement (Supplementary Fig. 14) and refer to it in the main text alongside the reference to the longterm mean circulation shown by Placke et al. (also in the discussion section).

Page 7, first column, lines 338->: see the comment above. On lines 346-347 it is stated that that the results in [28] show that in longer timescales the gyre is visible. Was the gyre not clearly visible after 6 days and if so, is there a reason to suspect that the model results were not reliable? What was the resolution of the ocean model? Can the grid resolution explain the lack of detail in the spatial distribution?

The word “resolved” was not the best choice to describe short-term (days) vs. long-term (months) model performance. The gyre is clearly visible in the model data also when looking at instantaneous data on 5 October (the horizontal resolution of the driving ocean currents was 1’ latitude x 1’40” longitude – in the revised manuscript, we add this information to Table 1 and Supplementary Table 1). However, as we describe, the exact placement may be off in the short-term. By contrast, over a timescale of months, the average placement of the gyre in the model corresponds well to the observations from Mohrmann et al., which is what the statement “*the gyre circulation and the broader methane plume is well resolved over longer timescales and larger spatial scales [28]*” referred to. To make this point clearer, we modify the text as follows:

Note that despite inaccuracies in the short term, ~~the~~ gyre circulation ~~and~~ modelled based on the surface currents we use [60] is able to reproduce the observed broader ~~methane~~ plume of dissolved methane is well-resolved over longer timescales and larger spatial scales [30].

Page 7, second column, lines 338-> : The exact form of the transfer velocity is not yet known, so there are several formulations for it. What was the reason for using a transfer velocity formulation that differs from the one used in M23? According to Figures S7 and S13, the transfer velocity is nearly twice as high for the wind speeds observed during the flight. If the only difference would be the vertical distribution, its role could be better evaluated.

Some aspects of our ocean model and M23 (now M24), including the different transfer velocity formulations, differ because the models were developed independently. More similarities in the setups would indeed have helped to better evaluate the role of the remaining differences between the ocean models. Therefore, we summarize the possible explanations for the discrepancies we find in our discussion. As we stated in response to a comment by Reviewer #2, we hope that our findings stimulate further investigation by the research community of the uncertainty of the emission estimates from the ocean models.

Page 8, first column, line 374. Could you explain why overestimation of transfer velocity would cause smaller emissions?

We add the explanation to that sentence:

..., overestimation of the gas transfer velocity (leading to higher outgassing rates prior to and hence lower concentrations in the water on the day of our observations) ...

Chapter 4.2

Page 10. first column, line 509. Referring to the Supplementary material chapter S2.1 and Figure S4: what might have been the reason for the large differences between the CAMS-CO₂ and measured CO₂ during flight 2 at low altitudes? See also the second comment on Chapter 2.2.1 above.

The offset between CO₂ observations and CAMS was mostly caused by a missing calibration of the observations. In the revised manuscript, the offsets of the now-calibrated CO₂ observations are much closer during flight 2 and similar to those of flight 1. Note that, while the calibration shifted both measured CO₂ and CH₄, it only had minimal effects on our retrieved emissions and no effect on the discussion. Since the offset is not there anymore (and because CAMS offsets have very little effect on

our results in the first place), we refrain from adding more discussion on differences between CAMS and the observations in the manuscript.

Chapter 4.3

Page 11, first column, lines 511-514: when defining the footprint of the observed fluxes, was the atmospheric stratification taken into account or was it assumed to be neutral? The stratification has a strong impact on the location from where the flux originates. This can be important since the ocean model gave larger spatial distribution.

Yes, the transport model STILT does take into account the atmospheric stratification. It does so by parameterizing turbulence based on meteorological fields (here, from the WRF model) such as wind and virtual potential temperature. For the sake of brevity, we do not include such details on STILT in the manuscript, but refer the reader to the cited reference instead (Lin et al. 2003, <https://doi.org/10.1029/2002jd003161>)

Page 11, first column, lines 554, 557: what do words "nudge towards" mean? A more precise formulation would be helpful.

We added an explanation at the first occurrence of "nudge":

To ensure consistency with weather observations, both WRF runs use grid nudging towards ERA5 in the outer domain. Grid nudging means non-physical tendency terms are applied in the meteorological simulation that keep the deviation from ERA5 small.

Page 11. second column, line 511: what drift needed to be avoided?

We clarified the sentence as follows:

Old:

To avoid drift in the inner domain, Meteo B started on 4 October 18:00 UTC.

Revised:

Since our observations only cover 5 October, Meteo B was started much later, on 4 October 18:00 UTC, to avoid drift of the simulated meteorological fields in the inner domain away from observed meteorological conditions.

Chapter 4.4

Page 11, second column, lines 553-560: is it realistic to assume that the mean prior flux estimate is zero? How much would the mentioned use of ocean model results (line 560) change the situation?

Indeed, the zero mean prior flux has a bias. However, atmospheric enhancements modeled using the posterior emissions had no mean bias (Table 2). Hence, in our setup, the bias of the prior does not degrade the posterior emissions in the area to which our observations are sensitive. In other words, the data constraint is strong enough to alleviate the bias of the prior flux. As discussed in the manuscript, there is a possibility that there were emission outside of that area, and an unbiased prior flux estimate would be necessary for an accurate posterior flux in that area. Therefore, we initially planned to use the ocean model as prior flux to alleviate any potential insufficient spatial coverage of our measurements.

The ocean model does indeed place emissions to the west of the northern leak sites (Fig. 3b), where our observations were not sensitive (Fig. 2). However, as we argue in the paper, the atmospheric observations reveal that this band of emissions was in reality extending to the southwest instead. Therefore, using the ocean model as prior flux would likely have retained a wrong emission pattern in the north of the domain, and thus degraded the inversion result.

We add a summary of this information to the methods:

Ideally, an inverse model features an unbiased prior flux estimate. However, the spatial distribution of the fluxes from the ocean model is at odds with the atmospheric data (Fig. 3d). Therefore, we provide an independent emission estimate by setting the mean prior flux estimate to zero throughout the domain instead of using ocean model results as prior flux estimate (note that despite the bias in the prior flux towards zero, atmospheric enhancements modelled using the posterior flux are unbiased; Table 2).

Chapter S1.2.1

Since the spatial distribution given by the ocean model is blurred compared to the atmospheric measured, could this indicate that the assumption of evenly distributed particles is not correct?

The initial distribution of the particles has little influence on the “blurriness” of the ocean model emissions. Instead, the “blurriness” is controlled by the horizontal diffusivity through the random velocity spread σ_i . We estimated σ_i via comparison with SOOP Finnmaid data. Thus, these data are ultimately what controls how large the emitting area in the ocean model is. Our Fig. S1 shows how diffusive the ocean model results are for different values of σ_i and compare them to the SOOP Finnmaid data. To make the minor role of the release radius clearer to the reader, we provide the maximum radius of the release (315 m) in the revised Supplementary Methods 3.1:

The mass of methane released per hour, the location of each of the six release points (two per leak), and the (time dependent) radius of the release into the mixed layer (maximum 315 m) are taken from the near-field model from D23 described above.

The rising gas bubbles caused mixing and turbulence. The modelled mixed layer depth was at a depth of 20-25 m (Figure S11): how reasonable is the assumption that the methane was mixed instantly to the whole layer? That is, were the conditions such that the mixing layer was mixed at least to the depth of the mixed layer in the given time frame?

As explained in response to a comment by Reviewer #1, the assumption of fast mixing is best justified by the fact that observations shown in companion manuscript Mohrmann et al. (ref. [30] in the revised text) show no consistent depth gradients of the dissolved methane mixing ratio in the surface layer, confirming that it was indeed well mixed. In the revised manuscript, this observation-based argument replaces the model-based argument from the original submission (see response to Reviewer #1 for the edited text).

Was the radius of the circle describing the gas plume constant in the mixed layer in the model used?

The surface spread radius was constant in depth, since our ocean model does not have any variability in the vertical dimension. Temporal variations were dynamically determined by the fountain and surface spread model from Dissanayake et al. (2023), described in Sect. S2.2 therein. To make this clear, we

add that the radius was “time dependent” here (see the response above to the first comment on Sect. S1.2.1).

Minor issues

According to the reference list, references [11] and [28] will be/are submitted to the same journal. It might be good to mention this in the outline of the manuscript in Chapter 1.

To stay consistent with the companion manuscripts, which do not mention the common submission, we decided to make no special mentions of these manuscripts, beyond citing them where they intersect with our work.

Chapter 4.2

Page 10, first column, line 482: above a background level?

Changed to “*above the background level*”.

Reviewer #3 (Remarks on code availability):

The work consists of multiple parts and rather heavy calculations: the codes should be run on a supercomputer which was not possible for this reviewer. The readme-files were carefully written and at least superficially judged usable for potential user. Also the arrangement of the material was clear. The reference to Wanninkhof 2014 was missing from the main readme-file under Ocean model.

We would like to thank Reviewer #3 for their constructive comments!